# Tff3 Deficiency Differentially Affects the Morphology of Male and Female Intestines in a Long-Term High-Fat-Diet-Fed Mouse Model

**DOI:** 10.3390/ijms242216342

**Published:** 2023-11-15

**Authors:** Kate Šešelja, Iva Bazina, Milka Vrecl, Jessica Farger, Martin Schicht, Friedrich Paulsen, Mirela Baus Lončar, Tatjana Pirman

**Affiliations:** 1Department of Molecular Medicine, Ruđer Bošković Institute, Bjenička 54, 10 000 Zagreb, Croatia; seseljakate@gmail.com (K.Š.); ivab1606@gmail.com (I.B.); 2Institute of Preclinical Sciences, Veterinary Faculty, University of Ljubljana, Gerbičeva 60, 1000 Ljubljana, Slovenia; milka.vrecl@vf.uni-lj.si; 3Institute of Functional and Clinical Anatomy, Faculty of Medicine, Friedrich-Alexander-University Erlangen-Nürnberg, 91054 Erlangen, Germany; jessi.farger@gmail.com (J.F.); martin.schicht@fau.de (M.S.); friedrich.paulsen@fau.de (F.P.); 4Department of Animal Science, Biotechnical Faculty, University of Ljubljana, Jamnikarjeva 101, 1000 Ljubljana, Slovenia

**Keywords:** trefoil peptide 3, duodenum, cecum, high-fat diet, oxidative stress, ER stress, apoptosis, inflammation

## Abstract

Trefoil factor family protein 3 (Tff3) protects the gastrointestinal mucosa and has a complex mode of action in different tissues. Here, we aimed to determine the effect of Tff3 deficiency on intestinal tissues in a long-term high-fat-diet (HFD)-fed model. A novel congenic strain without additional metabolically relevant mutations (*Tff3*-/-/C57Bl6NCrl strain, male and female) was used. Wild type (Wt) and Tff3-deficient mice of both sexes were fed a HFD for 36 weeks. Long-term feeding of a HFD induces different effects on the intestinal structure of Tff3-deficient male and female mice. For the first time, we found sex-specific differences in duodenal morphology. HFD feeding reduced microvilli height in Tff3-deficient females compared to that in Wt females, suggesting a possible effect on microvillar actin filament dynamics. These changes could not be attributed to genes involved in ER and oxidative stress, apoptosis, or inflammation. Tff3-deficient males exhibited a reduced cecal crypt depth compared to that of Wt males, but this was not the case in females. Microbiome-related short-chain fatty acid content was not affected by Tff3 deficiency in HFD-fed male or female mice. Sex-related differences due to Tff3 deficiency imply the need to consider both sexes in future studies on the role of Tff in intestinal function.

## 1. Introduction

Trefoil factor family protein 3 (Tff3) is a small peptide (59 amino acids; 7 kDa) that is a member of the trefoil factor family proteins (Tffs), comprising the Tff1, Tff2, and Tff3 proteins [1]. Tffs share a conserved motif of a well-defined three-loop structure, reminiscent of a trefoil stabilized by three disulfide bonds. Tffs are predominantly expressed in the gastrointestinal tract, where they protect the mucosal surface [2]. The presence of Tff3 in the bloodstream and various other organs, including the liver, brain, pancreas, and lymphoid tissue, indicates its general importance in organisms [3,4]. The mode of action of Tffs ranges from a simple increase in mucous viscosity, cytoprotection, and antiapoptotic and chemotactic effects to more complex immune regulatory functions [5,6]. Tff expression patterns are altered in various tumors, implying a strong association with tumorigenesis [7].

Tffs were long thought to be involved in mucosal repair by promoting cell migration (“restitution”) via their weak chemotactic and anti-apoptotic effects. Recent data on the molecular forms of human Tffs have revealed a more complex situation. Tff1 and Tff3 occur in vivo in different molecular forms and can form disulfide-linked heterodimers. In the intestine, Tff3 occurs as a Tff3-FCGBP (Fc fragment of the IgG binding protein (FCGBP)) heterodimer, Tff3-Tff3 homodimer, and Tff3 monomer [8,9,10,11]. This indicates that the biological functions of Tff peptides are complex [12].

Additionally, minimal amounts of Tff peptides are secreted in an endocrine manner, for example, by the central nervous system (CNS) [12] and immune system [13,14]. The potential role of the *Tff3* gene in a metabolism-related condition called diabesity (a combination of diabetes and obesity) was first revealed by quantitative trait locus (QTL) analysis of a mouse model of diabetes, the Tally Ho mouse strain [14], in which the complete reduction of Tff3 in the liver was the most dramatic change in early diabetes. Regulation of Tff3 with appropriate food intake and improvement in glucose tolerance in a diet-induced obesity model raises additional questions regarding the involvement of Tff3 in metabolic pathways [15]. This raises the question of whether Tff3 participates in the gut–liver–brain (GLB) axis. Animal mouse models of Tff protein deficiency are crucial for elucidating the physiological functions of specific proteins. Tff3-deficient mice show markedly increased sensitivity in a dextran sulfate sodium (DSS) colitis model, and these animals are particularly sensitive to radiation-induced mucosal injury and chemotherapy [16,17]. Tff3-deficient mice with a mixed genetic background (sv129/C57BL/6J) exhibit altered liver lipid metabolism [18]. It is well known that the phenotype of a given single gene mutation in mice is modulated by the genetic background of the inbred strain in which the mutation is maintained. This effect is attributable to the modifier genes, which act in combination with the causative gene [19]. A previously existing Tff3-deficient mouse strain had a mixed [18] or C57BL/6J background [16]. C57BL/6J mice have additional genetic variations in the nicotinamide nucleotide transhydrogenase (Nnt) gene, which encodes a mitochondrial redox-induced proton pump that links NADPH synthesis to the mitochondrial metabolic pathway. The Nnt mutation itself modulates metabolism [20,21] and the immune response [22] and hides the physiological function of the investigated candidate protein. The mixed genetic background poses the problem of ensuring a proper wild-type (Wt) control group due to random combinations of genetic variations [23]. Using the *Tff3*-/-//C57BL/6N mouse model, we overcame the issues of mixed genetic backgrounds and additional mutations in the C57BL/6J strain. Since the Tff3 protein is present in the blood circulation, intestine, liver, and brain, its role in these organs and the pathology of the GLB axis is of great interest.

This study aimed to determine the effect of Tff3 deficiency on the intestines of male and female mice under metabolically relevant conditions of long-term consumption of a high-fat diet (HFD) known to have pathological effects on the GLB axis. To exclude the effects of additional mutations and mixed genetic variations, we used a new congenic *Tff3*-/- strain on the C57BL/6NCrl genetic background with no additional metabolism-relevant mutations [24]. As existing data link Tff3 to metabolism, it is of great interest to investigate the systemic role of Tff3 deficiency in long-term HFD feeding (36 weeks). In this study, we monitored the effects of long-term HFD feeding on intestinal morphology and microbiome-relevant short-chain fatty acid (SCFA) content. The effect of Tff3 deficiency on the expression of genes involved in Tff3 function-related pathways (oxidative and endoplasmic reticulum (ER) stress, apoptosis, and inflammation) was determined using quantitative PCR (qPCR). Since Tff3 is regulated by estrogens [25] and metabolic events are influenced by sex, we monitored the impact of long-term HFD consumption on Wt and Tff3-deficient male and female mice.

## 2. Results

### 2.1. Body Weight and Glucose Level

Body weight and glucose levels, as markers of general health status, were determined before the mice were sacrificed at 36 weeks after 32 weeks of HFD feeding. After long-term HFD consumption, the body weights were similar in all four animal groups (Figure 1a). There were no sex differences between the specific genotypes. The statistically significant effect on glucose level (Figure 1b) was a sex-related difference in the *Tff3*-/- strain. *Tff3*-/- female mice had significantly (*p* < 0.01) lower blood glucose levels than *Tff3*-/- male mice at the age of 40 weeks, after 36 weeks of HFD feeding.

### 2.2. Tff3 Deficiency Affects Intestinal Morphology in the HFD Model

Villus height and crypt depth were measured as indicated (Figure 2). The longest villi of the duodenum were in the Wt female group, and the smallest villi were in the *Tff3*-/- female group (Table 1). The crypt depth of all four groups is shown (Table 1); there was a significant (*p* = 0.0216) difference among the different male genotypes. The ratio between the height of the villi and the depth of the crypt was the widest in the Wt female group (7.3:1) and narrowest in the Wt male and *Tff3*-/- female groups (5.9:1). The *Tff3*-/- male group was in the middle (6.5:1). There were significant differences among different female genotypes (*p* = 0.0224) and among Wt males and females (*p* = 0.0301); however, in the male groups, the differences were not significant. Differences in crypt depth in the cecum were significantly greater in the female group than in the male group of the same genotype (Table 1).

The ratio of villus height to average diameter was not significantly different among the groups (Table 1). Representative HE-stained histological sections are shown in the Appendix A. Staining with Alcian blue (PAS), which is used to distinguish neutral and acidic mucin in goblet cells, showed no obvious differences between the groups studied. In the duodenum, goblet cells containing a mixture of neutral and acidic mucins were predominant, with the exception of individual goblet cells with neutral mucins located at the base of the villi and crypt of Lieberkhün (PAS-positive, magenta stained). In the cecum, goblet cells were more numerous, and those at the base of the crypt were predominantly Alcian blue-positive (stained blue; acidic mucins). Goblet cells with neutral PAS-positive mucin were located near the epithelial surface of the upper crypt. Goblet cells containing a mixture of neutral and acidic mucins (stained blue–purple) were located along the crypt (Appendix A).

In contrast, there were no differences in villi height or crypt depth in either the duodenum or the cecum of mice fed the standard diet (SD) (Appendix A). There was no effect of sex or genotype, but mice fed the SD had significantly (*p* < 0.05) increased duodenal crypt depth compared to that of mice fed a HFD (Appendix A). In contrast, duodenal villi were significantly smaller in female Wt and male *Tff3*-/- mice (Appendix A). *Tff3*-/- males fed the SD had significantly deeper cecum crypts, which is the consequence of the reduced crypt depth due to a HFD (Appendix A). The opposite result was obtained in both female lines. Shallow crypts were observed when mice were fed the SD, and the crypt depth of both female groups on the HFD was deeper than that of males.

Transmission electron microscopy (TEM) analysis revealed no major pathological changes in any of the four animal groups fed a long-term HFD (Figure 3). The lamina propria of *Tff3*-/- female mice appeared to be jagged and loose. However, the same was true for the Wt male mice (*cf.* Figure 3a,d). *Tff3*-/- male mice had villi heights similar to those of Wt male mice (Figure 3a,b). The most apparent morphological alteration was observed in the microvilli of *Tff3*-/- female mice. In female *Tff3*-/- mice (Figure 3c,d), the microvilli height was reduced by 30% compared to that of female Wt mice (Figure 4; height mean value Wt 1.8 µm, *Tff3*-/- 1.3 µm). More lipid inclusions were observed in the enterocyte cytoplasm of Wt male mice (Figure 3a).

### 2.3. SCFA Content as an Indicator of the Microbiome

We monitored the content of several SCFAs in the cecum to determine possible microbiome differences between the monitored groups (Table 2). The levels of acetic acid, propionic acid, n-and i-butyric acid, and i-valerianic acid in the cecum were not affected by genotype or sex, and there was a high variability in the determined values within the groups.

### 2.4. Expression of Genes Involved in Disease-Relevant Pathways

The impact of Tff3 deficiency on Tff3 function-related pathways (oxidative and ER stress, apoptosis, and inflammation) was determined using qPCR (Figure 5, Figure 6, Figure 7 and Figure 8). The expression of numerous genes involved in ER stress (Figure 5), oxidative stress (Figure 6), inflammatory pathways (Figure 7), and apoptosis (Figure 8) was determined in the duodenum of all four mouse groups. HFD feeding for 36 weeks resulted in significantly increased expression levels of the spliced X-box-binding protein 1 (*sXbp1*) gene in Tff3-deficient animals compared to those in their Wt counterparts. This was true for both male and female mice (Figure 7a,b). No significant differences were observed between males and females of the same genotype (Figure 7c,d).

## 3. Discussion

An improper diet rich in saturated fat and sugars is a risk factor for the majority of diseases in the Western world, including human inflammatory bowel diseases and metabolic diseases, such as liver steatosis and diabetes, leading to cardiovascular diseases [26]. It even has an impact on cognitive function and the onset of neurodegenerative diseases [27,28]. A HFD and overnutrition lead to intestinal dysbiosis, an increase in the populations of gram-negative bacteria and lipopolysaccharide (LPS) content, and induction of proinflammatory cytokines that lead to disturbance in the intestinal barrier, the so-called “leaky gut”, which additionally disrupts the gut–liver axis [29]. Chronic intestinal inflammation leads to systemic inflammation coupled with an increase in oxidative stress in different tissues, leading to metabolic disturbances and the development of obesity. The interconnection between exaggerated inflammation and oxidative stress caused by overnutrition contributes to cancer development [29]. The Tff3 protein, as part of the intestinal mucus, participates in the protection of intestinal mucosal integrity by promoting intestinal restitution, which involves its weak mutagenic function, antiapoptotic function, and angiogenic effects through its lectin activity [30]. Currently, the following transmembrane proteins have been reported to have a binding affinity for the Tff3 peptide: integrin [31], CRP-ductin, also known as gp-340/deleted in malignant brain tumors 1 (DMBT1gp340) [31,32], C-X-C chemokine receptor type 4 (CXCR4) and seven transmembrane-spanning receptor (CXCR7) [33,34], proteinase-activated receptor 2 (PAR2) [35], and leucine-rich repeat and Ig domain containing 2 (LINGO2) [36]. Additionally, Tff is linked to inflammation [6] and is regulated by different components of the immunomodulatory pathway [37,38]. More recently, it has been recognized that Tff3 also affects liver metabolism [39], making it important to elucidate its role in the gut–liver axis. Tff3 is present in the blood circulation, liver, and pancreas and is regulated by food intake. Additionally, Tff3 improved the glucose tolerance in a diet-induced obesity model, raising the question of the involvement of Tff3 in metabolic pathways. We monitored the effect of Tff3 deficiency on the intestines of male and female mice fed a long-term HFD, which mimics the physiological events in the development of different diseases. A long-term HFD disrupts mucosal integrity and increases intestinal inflammation, oxidative stress, and ER stress [40]. A HFD contributes to intestinal damage by reducing mucin 2 and Tff3 expression, which in turn affects mucus quality [40,41]. *Tff3*-/- mice were more susceptible to DSS-induced intestinal damage [16]. The importance of Tff3 in intestinal mucus was additionally indicated in *Arhgap17*-/- [42] and *Tlr2*-/- [43] mice. These mice also showed reduced expression of intestinal Tff3, which strongly increased the severity of DSS-induced colitis. Rats overexpressing human Tff3 had a reduced extent of indomethacin-induced jejunal injury because they showed a 29% reduction in villi height versus a 51% reduction in those of the controls [44]. The novel congenic mouse model *Tff3*-/-/C57BL/6NCrl, without additional mutations present in the C57Bl/6J strain and without the impact of unintentional genetic variants due to a mixed genetic background, was fed a HFD to determine the impact of Tff3 deficiency and Tff3 protein function in this disease-related model that appears to include some adaptation mechanisms [45]. We found that Tff3-deficient males (21 weeks old) had significantly reduced body weight, which could not be noticed at later stages (age 36 weeks and 40 weeks) of HFD feeding [46]. Tff3-deficient females exhibited no such body weight time line variation, but at the end point (40 weeks), they had significantly lower blood glucose levels (Figure 1) [46].

Interestingly, monitoring the effect of Tff3 deficiency on the intestinal morphology of 40-week-old mice (upon 36 weeks of HFD) revealed sex-related differences (Table 1 and Figure 2). Crypt depth values indicate the proliferative activity of epithelial cells [47], whereas lower crypt depth values suggest a reduction in the metabolic cost of intestinal epithelium turnover [48].

HFD-fed Tff3-deficient mice did not show a change in intracellular connections compared to WT mice, but microvilli height was affected (Figure 3). The apical surface of intestinal absorptive cells is covered with numerous microvilli and well-ordered finger-like membrane protrusions supported by an organized bundle of actin filaments [49,50] that settle on a complex terminal web area. Together, they form the brush border (BB), which is a crucial dynamic interface specialized for the secretion and absorption of salts and nutrients that modulate gut homeostasis. BB assembly and dynamics are precisely regulated [51]. The microvillar core is composed of polarized actin filaments with different association and/or dissociation rates at both extremities of each microfilament [52]. The addition of monomeric actin occurs exclusively at the tip of the microvillus; actin disassembles at the pointed end, suggesting that actin polymerization drives microvillus formation [52,53]. This mechanism results in a dynamic equilibrium between polymerization and depolymerization that maintains the microfilament at a constant length. However, when unbalanced, it can induce growth or shrinkage of the actin microfilament [54], but currently, the exact mechanism is unknown.

Cytoskeletal changes are regulated by Rho family GTPases, which initiate cell migration [55]. Rho triggers the formation of contractile stress fibers and focal adhesion complexes, and the Rho family member Rac induces rapid actin polymerization and lamellipoid protrusions [56,57]. Several studies have shown that Tff peptides can activate Rho family members. Tff stimulation in vitro in the presence of active Src kinase promotes Src localization to focal adhesions and activates Rho family members [58].

Additionally, mice deficient in ARHGAP17 (Rho GTPase Activating Protein 17) have markedly reduced expression levels of intestinal Tff3, which affects the severity of intestinal injury induced by DSS [42]. In vitro experiments using premalignant colonic epithelial cells showed that Tff peptides require activated Src/RhoA to induce cellular invasion [58,59]. Tff-mediated actin cytoskeleton rearrangement has been demonstrated in Caco-2 cells, wherein Tff3 is capable of restricting platelet-activating factor-induced disruption of the F-actin cytoskeleton [60]. These studies highlighted the role of Tff in activating changes in the actin cytoskeleton during cell migration. Proteomic analysis of the effects of a HFD on the mouse small intestine revealed changes in different metabolic and immune defense-related proteins [61]. Interestingly, proteomic changes suggest that cellular adhesion and cell–cell interactions increase after a HFD, whereas the movements of the microvilli decrease. In our model, we cannot explain why long-term HFD feeding has sex-related effects on microvillar dynamics, resulting in shortened microvilli in Tff3-deficient female mice compared to that in Wt females fed a HFD. To exclude the possible impact of the microbiome, we analyzed the relevant SCFA contents in the cecum. Microbiome diversity is affected by HFD consumption [62]. Specifically, bacterial species that feed on nondigestible dietary fibers are important for intestinal function and produce metabolites that exert positive effects on the intestinal mucosa. Examples include SCFAs, mainly acetate, propionate and butyrate. Butyrate is the primary energy source for colonocytes and maintains intestinal homeostasis through anti-inflammatory actions [63,64]. Tff3 is linked to the intestinal innate immune response, as its expression is induced after the activation of Toll-like receptor 2 (TLR2) by commensal bacteria [43]. At the cellular level, SCFAs can have direct or indirect effects on processes such as cell proliferation, differentiation, and gene expression [65].

To elucidate the possible molecular changes in the duodenum of Wt and Tff3-deficient mice of both sexes, we determined the expression of numerous genes involved in disease-related pathways, including ER and oxidative stress-related, inflammatory, and apoptosis pathways. The predominant intestinal Tff3 form in humans (and probably in mice) is a disulfide-linked heteromer with the IgG FCGBP [9,10]. The precise function of FCGBP and its heteromers with Tff3 (Tff3–FCGBP) is still unknown; however, it includes some of the previously described effects of the Tff3 mode of action. There are indications that FCGBP plays a role in the innate immunity of the mucous epithelia, which are thought to regulate pathogen attachment and disease progression [66]. In the prolonged HFD model, we did not find any statistically significant impact of Tff3 deficiency on the expression of inflammation-related genes, either oxidative stress- or apoptosis-related genes, although the expression levels of most of these genes were upregulated. Interestingly, among the numerous genes involved in ER and oxidative stress, inflammation, and apoptosis pathways, the only statistically relevant change in the duodenum of Tff3-deficient mice in both sexes was the increased expression of spliced X-box binding protein 1 (sXBP1), the key transcription factor that promotes the adaptive unfolded protein response (UPR) (Figure 5). It is known that long-term HFD feeding disrupts lipid metabolism and causes inflammation in adult male rats through ER stress [40].

The ER is the main cellular site for the synthesis and processing of proteins, lipids, and carbohydrates. Several conditions, such as disturbed lipid homeostasis, disturbed calcium signaling, oxidative stress, inhibition of glycosylation, and increased protein synthesis, can affect ER function. The accumulation of defective proteins in the ER and decreased ER-associated degradation can cause ER stress and activate the UPR, a complex network of adaptive responses that restores ER function [67].

Disruption of ER homeostasis initiates the UPR through three canonical signaling pathways: the ERN1/IRE1 to nuclear signaling branch; the EIF2AK3/PERK (eukaryotic translation initiation factor 2 alpha kinase 3) branch; and the ATF6 (activating transcription factor 6) branch. Through these branches, the UPR controls a complex network of adaptive responses to restore normal ER function. Activation of ERN1, the most conserved UPR regulator, initiates unconventional splicing of the mRNA that encodes the transcription factor XBP1, producing sXBP1, which enters the nucleus and regulates the expression of UPR-related genes [68,69].

UPR activation was monitored by qPCR analysis of downstream targets of the UPR. The analyzed genes were common genes used to examine UPR activation: endoplasmic reticulum chaperone BiP (BiP), activating transcription factor 4 (ATF4), DNA damage-inducible transcript 3 (CHOP), ER degradation-enhancing alpha-mannosidase-like protein 1 (EDEM1), heat shock protein 90 (GRP94), and sXBP1 [70]. Upon perturbation of protein folding, BiP dissociates from the three key master regulators of the UPR, IRE1, pancreatic EIF2-α kinase (PERK), and ATF6, and consequently activates them. ATF4 and CHOP are transcription factors activated in the PERK signaling pathway [71,72]. In response to ER stress, IRE1 cleaves XBP1, resulting in the expression of sXBP1, an active UPR transcription factor that exerts strong pro-survival effects under various conditions [73]. EDEM1 is an ER-associated protein degradation (ERAD) chaperone [74], and GRP94 performs unique chaperone functions in the ER [75]. The primary purpose of the UPR is to enhance protein degradation, reduce protein synthesis, increase ER protein-folding capacity, and upregulate chaperones needed for protein folding; therefore, it is clear that the UPR plays a crucial role in tissue homeostasis. Interestingly, in the duodenum of HFD-fed Tff3-deficient mice (male and female), the expression of only sXBP1 was significantly increased. In response to ER stress, XBP1 is processed into its spliced form, sXBP1, which produces a transcription factor that binds to a promoter element referred to as an X-box. sXBP1 reestablishes ER homeostasis and regulates glucose and lipid metabolism [76,77].

Unresolved ER stress is the primary contributor to the pathogenesis of inflammatory bowel disease (IBD). Studies have also suggested that ER stress is the primary cause and/or consequence of inflammation [78]. Tff3-deficient mice fed a HFD did not show an increased inflammatory response, most likely due to increased activation of sXBP1, which maintains tissue homeostasis.

This manuscript presents part of the data from extensive research on the systemic impact of Tff3 deficiency on the gut–liver–brain axis. In our research, we used prolonged feeding of a high-fat diet to mimic physiologically relevant conditions, including aging, contributing to different diseases in that axis. Here, we present the impact of Tff3 deficiency on the intestine in both sexes. To our surprise, the major intestinal finding was a reduced surface area of the intestine in Tff3-deficient female mice fed a HFD (reduced duodenal villi and microvilli height) for 36 weeks. This was not observed in Tff3-deficient female mice fed a standard diet for the same duration. During the monitored period from 4 to 40 weeks of age, we performed functional metabolic tests of glucose and insulin tolerance, as presented elsewhere [46]. We observed different time-related changes but no major difference in body weight due to Tff3 deficiency. One of the possible reasons for the reduced intestinal surface in Tff3-deficient females fed a HFD could be related to the feeding behavior or food absorption abilities of mice. Unfortunately, the data collected by the research concept presented here cannot resolve this issue. Additional experiments including enzymatic activities in the brush border are needed to further address whether feeding behavior is the cause of the observed morphological changes in the intestines of Tff3-deficient females fed a HFD.

## 4. Materials and Methods

### 4.1. Animals

The Tff3-deficient mouse strain on the C57BL/6NCrl (Charles River) background was developed from an existing Tff3-deficient mixed background strain (C57BL/6J/SV129) using a ‘speed congenics’ approach as described [24]. Tff3-deficient mice (*Tff3*-/-/C57BL/6NCrl) and the appropriate wild-type strain C57BL/6NCrl were raised in the animal facility of the Ruđer Bošković Institute under standard care conditions. Female and male wild-type (C57BL/6NCrl) and Tff3-deficient genotype (*Tff3*-/-//C57BL/6NCrl) mice were monitored daily. A group of animals consisting of 10 male WT, 10 male *Tff3*-/-, 10 female WT, and 10 female *Tff3*-/- mice was fed a high-fat diet (HFD) from weaning at 4 weeks of age until sacrifice 36 weeks later (at 40 weeks of age). During that time, mice were monitored by functional metabolic tests of glucose tolerance (21 and 36 weeks old) and insulin tolerance (23 and 38 weeks old) as described in a previously published manuscript [46]. Weight and blood glucose levels at 40 weeks of age were determined. Five animals per group were used for the morphological analysis, and 5 were used for fresh tissue collection. Total body perfusion through the heart (50 mL phosphate-buffered saline (PBS) and 50 mL 4% paraformaldehyde (PFA)) was applied, and tissue was collected for paraffin embedding or ultrastructure processing (immersed in indium tin oxide (ITO) buffer). The mice were maintained at 21 °C with 60% humidity and a light–dark cycle of 12 h light/dark starting at 7.00 a.m. Experimental animal manipulations and procedures performed in the course of the Croatian Science Foundation grant IP-06-2016-2717 were approved by the local ethical committee.

### 4.2. Diet

The mice (4 weeks old) were fed a special HFD containing 60 kJ% fat (Lard) (ssniff Spezialdiäten GmbH, Soest, Germany; cat. number. EF E15742-34) until the age of 40 weeks. The crude nutrient content included 24.4% crude protein, 34.6% crude fat, 6.0% crude fiber, 5.5% crude ash, 0.1% starch, and 9.4% sugar.

### 4.3. Chemical Composition of the Diet

During the experiment, diet samples were collected for proximate analysis (Appendix A). Proximate analysis was performed in the laboratory by standard procedures [79]. Dry matter was oven dried at 95–100 °C (AOAC Official method 934.01), crude protein was determined by the copper catalyst Kjeldahl method (AOAC Official method 984.13), total fat was determined by extraction in petroleum ether (AOAC Official method 920.39), crude fiber was determined by the fritted glass crucible method (AOAC Official method 978.10), and crude ash content was determined by placing a certain amount of sample in a muffle furnace at 550 °C, followed by determination of the final weight (AOAC Official method 942.05). The content of macrominerals (Ca, P, Mg, K, and Na) was determined after ashing and preparation of an acid extract using atomic absorption spectrometry (1100 B; PerkinElmer Inc., Waltham, MA, USA).

### 4.4. Tissue Sampling, Staining, and Histological Measurements of Intestinal Tissues

After total body perfusion, the intestinal tract was removed and divided into the small intestine, colon, and cecum. One-centimeter-long portions of the small intestine and cecum were obtained from the following locations: the duodenum, one cm after the junction with the stomach, the cecum, and one cm from the beginning of the cecum. Samples were fixed in 10% buffered formalin (Shandon Formal-Fixx, 10% neutral buffered formalin; Thermo Scientific GmbH, Vienna, Austria) and then dehydrated and embedded in paraffin blocks (Tissue-Tek^®^ TEC™ 5 Tissue Embedding Console System, Sakura Finetek Europe) according to a standard procedure. Subsequently, an evenly spaced series of 5 μm histological sections with a section interval of 50 μm were prepared using a sliding microtome (Leica SM2000R; Nussloch, Germany). They were stained with hematoxylin and eosin (HE) and applied to a coverslip using the Gemini AS automated slide stainer and ClearVue cover slipper (Thermo Fisher Scientific, Waltham, MA, USA). In addition, an Alcian Blue pH 2.5-Periodic acid–Schiff (PAS) staining kit (Merck KGaA, Darmstadt, Germany) was used to distinguish between neutral and acidic mucins in goblet cells. The neutral and acidic mucins in goblet cells were stained magenta and blue, respectively. A mixture of neutral and acidic mucins results in purple, magenta–purple, or blue–purple staining of the goblet cells.

Histomorphometric analysis was performed on HE-stained tissue sections using a Nikon Eclipse Ni-UM microscope equipped with a DS-Fi1 camera and Imaging Software NIS Elements BR 4.6 (Nikon Instruments Europe B.V., Badhoevedorp, The Netherlands) as previously described [80]. The villus height was measured from the tip to the crypt–villus junction, and the crypt depth was measured from the crypt–villus junction to the base, as shown in Figure 1. The average villus diameter was measured at 50% of the villus height, and the villi area was calculated using the software NIS Elements BR 4.6 (Nikon instruments Inc. 1300 Walt Whitman Road, Melville, NY 11747-3064 USA) after defining the area of each villus. Representative tissue sections are presented using Adobe Creative Cloud.

### 4.5. SCFA Analyses

The contents of the cecum were squeezed out by finger pressure, collected in Eppendorf tubes, and stored at −20 °C until SCFA analysis. The volatile fatty acid (VFA) concentrations in the cecum contents were determined by gas chromatography using an Agilent 6890A GC system equipped with an FID detector (Agilent, Santa Clara, CA, USA) and a DB-FASTWAX UI capillary column (30 m × 0.25 mm × 0.25 µm) (Agilent, USA). Before injection (2 µL diethyl ether extract, split 10:1), diethyl ether extracts were prepared using the method reported by Holdeman and Moore [81] with some modifications, as described by Pirman et al. [82].

### 4.6. Q-PCR Analysis

The duodenum was collected, snap-frozen in liquid nitrogen, and stored at −80 °C for further analysis. Total RNA was extracted using an RNeasy Mini Kit (Qiagen, Hilden, Germany) according to the manufacturer’s instructions. cDNA was synthesized using the High-Capacity cDNA Reverse Transcription Kit (Applied Biosystems, Dreieich, Germany). qPCR analysis was carried out on the StepOnePlus™ qPCR System (Applied Biosystems) using SYBR Green I (Invitrogen, Waltham, MA, USA) detection chemistry and specific primers at specified conditions (Appendix A). Gene expression levels were normalized to two stable housekeeping genes, β-actin (ACTβ) and β2-microglobulin (B2M), and the specificity of amplification was further confirmed by melting curve analysis. Data were analyzed using software REST-MCS (version 2; https://www.gene-quantification.de/rest-mcs.html) (2^−∆∆Ct^ method), and changes were represented as log2 (fold change) using stable reference genes (ACTβ and B2M).

### 4.7. Ultrastructure

The mice fed a HFD (wt and *Tff3*-/-//C57Bl6/NCrl) were perfused with 4% PFA, and tissues were immersion-fixed immediately after removal in ITO fixative. The livers were postfixed in OsO4 and dehydrated in a graded ethanol series, and whole tissues were embedded in EPON resin. Tissue sections (1 µm-thick) were cut using an ultramicrotome (Ultracut E; Reichert Jung, Vienna, Austria) and stained with toluidine blue. Toluidine blue-stained slides were examined under a Biorevo BZ-9000 microscope (Keyence, Neu-Isenburg, Germany). Ultrathin tissue sections were cut, stained with uranyl acetate and lead citrate, and viewed using a transmission electron microscope, JEM-1400Plus (JEOL (Germany) GmbH, Freising, Germany). The height of microvilli in the duodenum of *Tff3*-/- female mice was compared to that of Wt females. We analyzed 3 animals per group (biological replicate), and each animal was presented with 4 different sections (technical replicates). In each section, the height of at least 10 different microvilli was measured. Pictures a were analyzed using ImageJ software (https://imagej.net/ij/). Statistical evaluation (*t*-test) and presentation was performed using GraphPad Prism v. 7 (Dotmatics, Boston, MA, USA) (https://www.graphpad.com/features).

### 4.8. Statistical Analysis

The results were analyzed using the general linear model (GLM) procedures of the SAS/STAT module (SAS Institute Inc., Cary, NC, USA), and the differences were determined by Tukey–Kramer multiple comparison tests, considering the genotype as the main effect separately for male and female mice, if indicated otherwise. Least squares means (LSM) were computed and are shown in the results. Dispersion is expressed as the standard deviation of the mean (SEM). Statistical significance was set at *p* < 0.05.

## 5. Conclusions

Tff3-deficient mice fed a HFD did not exhibit dramatic effects compared with WT mice fed a HFD. Tff3-deficient males were protected from weight gain after 36 weeks of HFD consumption. Activation of ER stress through sXBP1 was observed; however, it did not increase the expression levels of genes involved in inflammation, apoptosis, or oxidative stress. For the first time, we detected a difference in duodenal morphology, as Tff3-deficient females on a HFD had greatly shortened microvilli, which may be related to the effects on microvilli actin filament dynamics. Cecal crypt depth in Tff3-deficient males was reduced. These novel findings imply the necessity of using both sexes in future research on the function of Tffs.

## Figures and Tables

**Figure 1 ijms-24-16342-f001:**
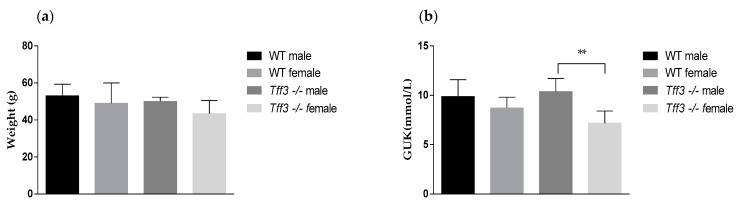
Effect of long-term HFD feeding on body weight and blood glucose levels in Wt and Tff3-deficient mice of both sexes (*n*~10 animals per group). (**a**) Bodyweights of Wt and *Tff3*-/- mice (male and female) and blood glucose levels (**b**) at the age of 40 weeks, after 36 weeks of HFD. ** *p* ≤ 0.01.

**Figure 2 ijms-24-16342-f002:**
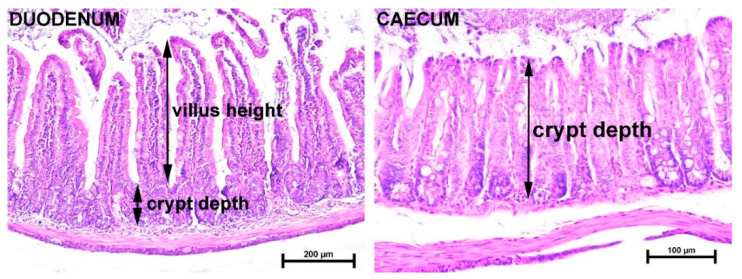
Photomicrographs of the duodenum and cecum. Villus height and crypt depth were measured as indicated. In the duodenum, villus height was measured from the tip of the villus to the crypt–villus junction, and crypt depth was measured from the crypt–villus junction to its base. Measurement of crypt depth is shown in the cecum. H&E staining; scale bars 200 µm (duodenum) and 100 µm (cecum).

**Figure 3 ijms-24-16342-f003:**
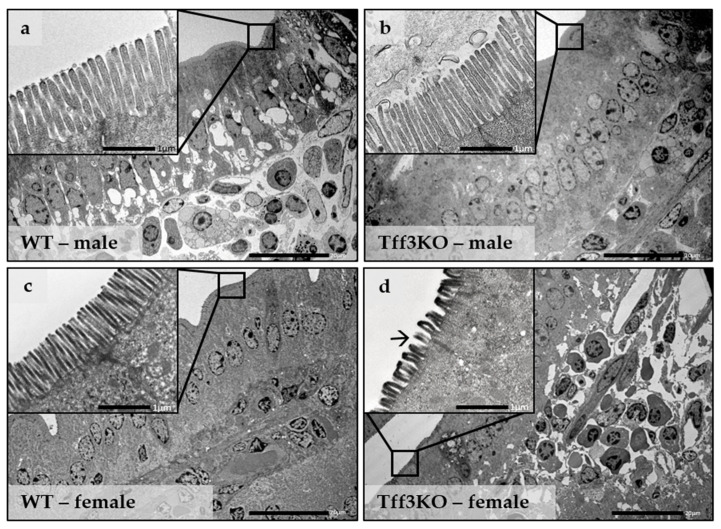
Duodenal ultrastructure of Wt and *Tff3*-/- mice fed a HFD. (**a**) Wt male; (**b**) *Tff3*-/- male; (**c**) Wt female; (**d**) *Tff3*-/- female. The height of the microvilli in the duodenum of *Tff3*-/- female mice is reduced (→); scale bars are 20 µm and 1 µm (inserts).

**Figure 4 ijms-24-16342-f004:**
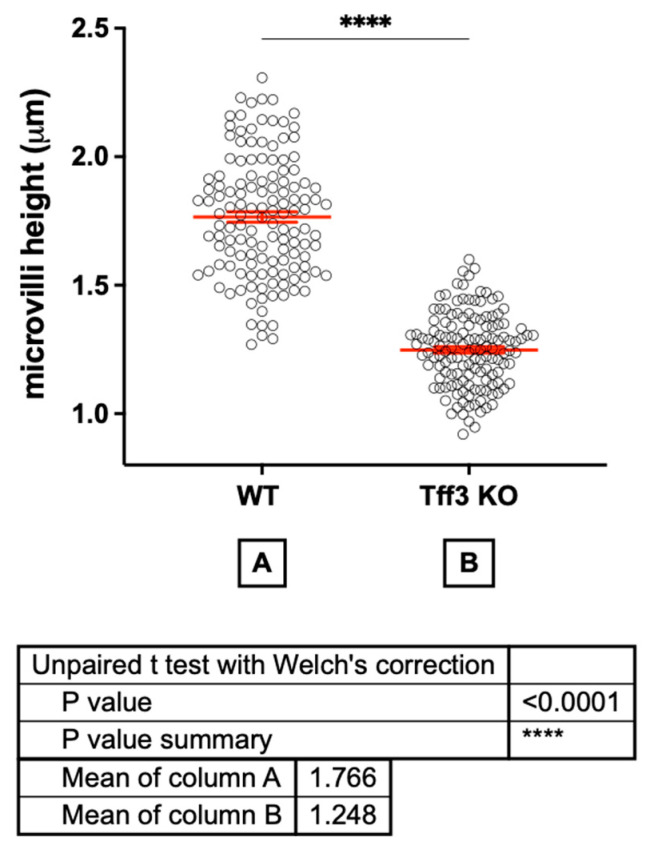
Height of microvilli in the duodenum of Wt and *Tff3*-/- female mice fed HFD. The microvilli in the duodenum of *Tff3*-/- female mice were significantly reduced. Compared with Wt females, Tff3-deficient females had 30% lower microvilli height. We analyzed 3 animals per group, and 4 different sections were presented for each animal. In each section, the height of at least 10 different microvilli was measured, red line is interval bar of SEM, **** *p* < 0.0001. Data were analyzed using ImageJ software (Version 1.50i, National Institute of Health, Bethesda, Md, USA) and *t*-test analysis.

**Figure 5 ijms-24-16342-f005:**
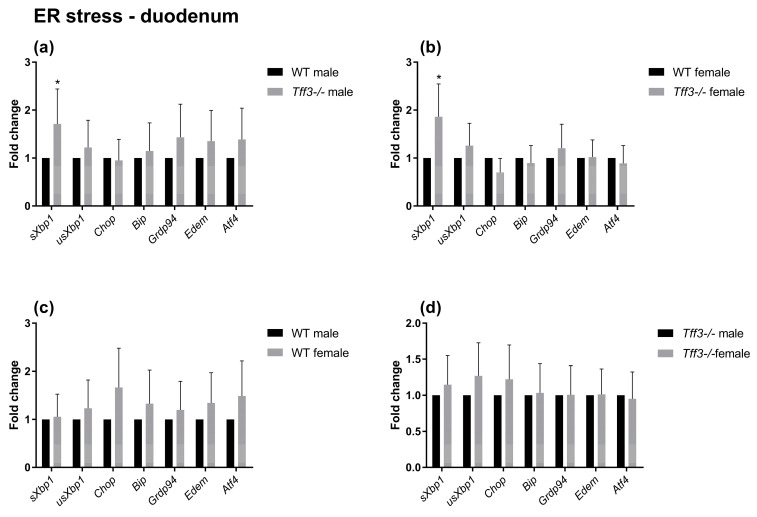
Effect of long-term HFD feeding on the expression of ER stress markers in the duodenum of Wt and Tff3-deficient mice of both sexes. We performed qPCR using SYBR green detection for each group (*n* = 5 animals per group). C_t_ data were analyzed by the REST program and are presented as fold change with upper limits. (**a**) The expression levels of genes in *Tff3*-/- male mice relative to those of the Wt male mice are shown; (**b**) The expression levels of genes in *Tff3*-/- female mice relative to those of the Wt female mice are shown; (**c**) Expression levels of genes in Wt female mice relative to those of the Wt male mice are shown; (**d**) The expression of genes in *Tff3*-/- female mice relative to that of the *Tff3*-/- male mice is shown. * *p* ≤ 0.05.

**Figure 6 ijms-24-16342-f006:**
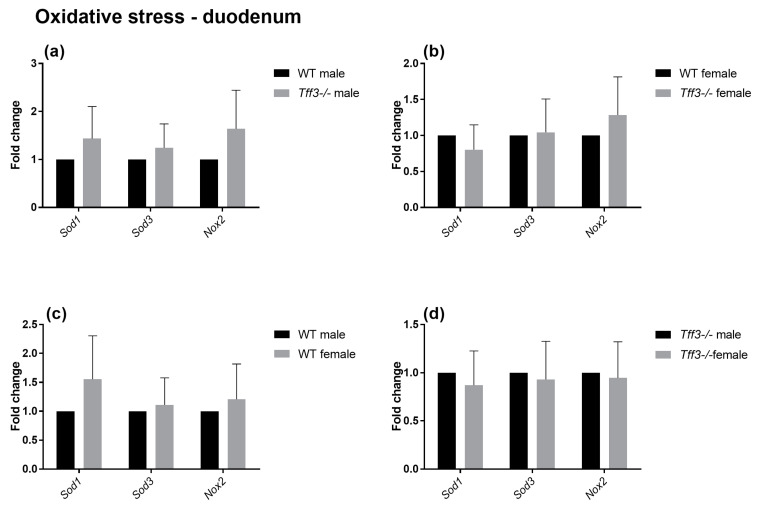
Effect of long-term HFD feeding on the expression of oxidative stress markers in the duodenum of Wt and Tff3-deficient mice of both sexes. We performed qPCR using SYBR green detection for each group (*n* = 5 animals per group). C_t_ data were analyzed by the REST program and are presented as fold change with upper limits. (**a**) The expression levels of genes in *Tff3*-/- male mice relative to those of the Wt male mice are shown; (**b**) The expression levels of genes in *Tff3*-/- female mice relative to those of the Wt female mice are shown; (**c**) Expression levels of genes in Wt female mice relative to those of the Wt male mice are shown; (**d**) The expression levels of genes in *Tff3*-/- female mice relative to those of the *Tff3*-/- male mice are shown.

**Figure 7 ijms-24-16342-f007:**
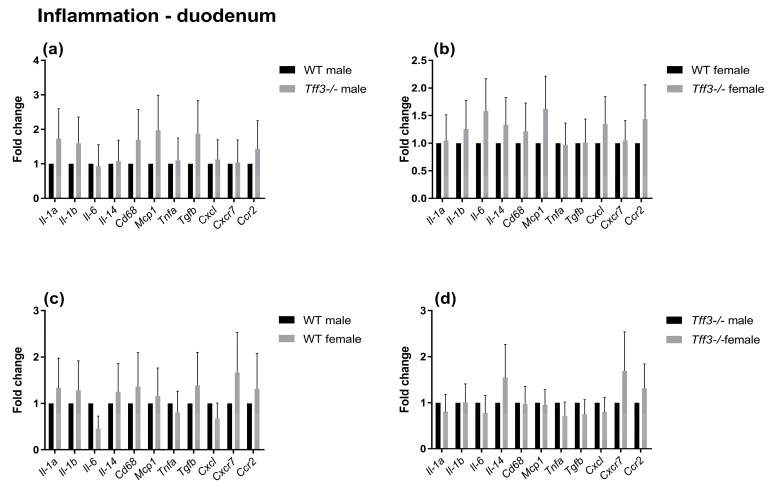
Effect of long-term HFD feeding on the expression of inflammation-relevant genes in the duodenum of Wt and Tff3-deficient mice of both sexes. We performed qPCR using SYBR green detection for each group (*n* = 5 animals per group). C_t_ data were analyzed by the REST program and are presented as fold change with upper limits. (**a**) Wt male mice; (**b**) The expression levels of genes in *Tff3*-/- female mice relative to those of the Wt female mice are shown; (**c**) Expression levels of genes in Wt female mice relative to those of the Wt male mice are shown; (**d**) The expression levels of genes in *Tff3*-/- female mice relative to those of the *Tff3*-/- male mice are shown.

**Figure 8 ijms-24-16342-f008:**
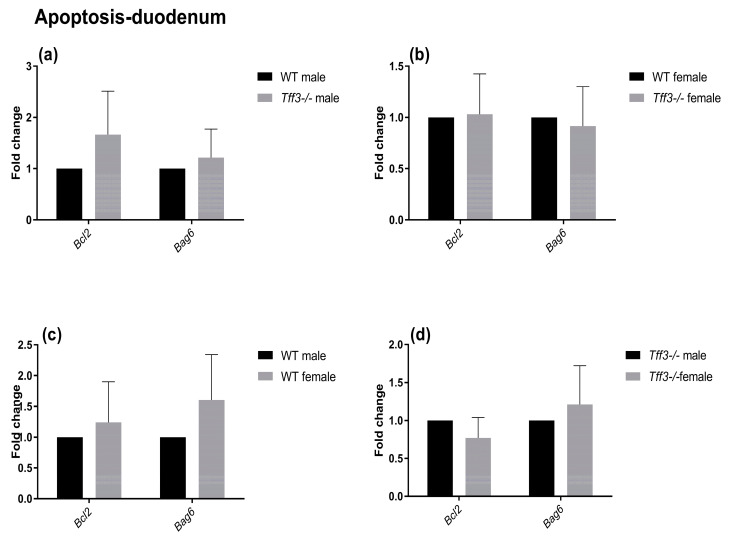
Effect of long-term high-fat diet treatment on the expression of apoptosis-related genes in the duodenum of Wt and Tff3-deficient mice of both sexes. We performed qPCR using SYBR green detection for each group (*n* = 5 animals per group). C_t_ data were analyzed by the REST program and are presented as fold change with upper limits. (**a**) The expression levels of genes in *Tff3*-/- male mice relative to those of the Wt male mice are shown; (**b**) The expression levels of genes in *Tff3*-/- female mice relative to those of the Wt female mice are shown; (**c**) Expression levels of genes in Wt female mice relative to those of the Wt male mice are shown; (**d**) The expression levels of genes in *Tff3*-/- female mice relative to those of the *Tff3*-/- male mice are shown.

**Table 1 ijms-24-16342-t001:** Histological measurements: height of the villi in the duodenum and depth of the crypt in the duodenum and cecum (average ± standard deviation) of HFD-fed mice.

Intestinal Histology	Wt Male (10)	*Tff3*-/- Male (10)	Wt Female (10)	*Tff3*-/- Female (10)
Duodenum				
Villi height (µm)	420.6 ± 83.2 *	501.6 ± 31.2 ^†^	531.7 ± 60.5	416.8 ± 33.6 ^§^
Villi average diameter (µm)	255 ± 31	278 ± 29	292 ± 26	249 ± 19 ^§^
Villi area (µm^2^)	57,321 ± 19,416	62,933 ± 1270	68,475 ± 1224	49,855 ± 7558 ^§^
Ratio villi height:diameter	1.64:1	1.81:1	1.82:1	1.68:1
Crypt depth	70.5 ± 5.0	77.1 ± 1.5 ^‡^	73.3 ± 6.0	71.7 ± 7.5
Ratio villi:crypt	5.9:1 *	6.5:1	7.3:1	5.9:1 ^§^
Cecum				
Crypt depth	166.2 ± 20.2 *	140.8 ± 12.0 ^†,‡^	213.9 ± 26.1	206.9 ± 17.1

The results are presented as the average ± standard deviation. Statistical significance was considered at *p* ≤ 0.05: *—Wt♂ vs. Wt ♀ (sex-related diff.) †—*Tff3*-/- ♂ vs. *Tff3*-/- ♀ (sex-related diff.) ‡—Wt♂ vs. *Tff3*-/- ♂ (gene-related diff.) §—Wt ♀ vs. *Tff3*-/- ♀ (gene-related diff.).

**Table 2 ijms-24-16342-t002:** Short-chain fatty acid concentration (g/kg sample) in the cecum (average ± standard deviation).

Short-Chain Fatty Acids	Wt Male (5)	*Tff3*-/- Male (5)	Wt Female (5)	*Tff3*-/- Female (5)
Cecum				
Acetic acid	1.80 ± 0.85	1.66 ± 0.46	1.99 ± 0.47	1.81 ± 0.41
Propionic acid	0.34 ± 0.12	0.38 ± 0.11	0.47 ± 0.10	0.42 ± 0.09
i-Butyric acid	0.04 ± 0.03	0.04 ± 0.02	0.04 ± 0.02	0.05 ± 0.01
n-Butyric acid	0.24 ± 0.15	0.29 ± 0.11	0.28 ± 0.09	0.30 ± 0.12
i-Valerianic acid	0.05 ± 0.03	0.06 ± 0.03	0.06 ± 0.02	0.06 ± 0.01
n-Valerianic acid	0.05 ± 0.03	0.05 ± 0.02	0.06 ± 0.02	0.06 ± 0.02
Sum	2.52 ± 1.13	2.49 ± 0.68	2.90 ± 0.57	2.70 ± 0.61
Ratio AA:PA:BA	5.33:1:0.80	4.77:1:0.90	4.31:1:0.68	4.33:1:082

The results are presented as the average ± standard deviation. AA, acetic acid; PA, propionic acid; BA, butyric acid.

## Data Availability

Data are contained within the article and Appendix A.

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
