# Peer review of "Tff3 Deficiency Differentially Affects the Morphology of Male and Female Intestines in a Long-Term High-Fat-Diet-Fed Mouse Model"

_ijms, 2023, doi:10.3390/ijms242216342_

Round 1
Reviewer 1 Report
Comments and Suggestions for Authors
- Basically this article tends to show that the effect of Tff3 deficiency leads to sex-related differences in duodenal microvilli upon prolonged exposure to high-fat diet. The findings are potentially interesting but very preliminary This article is the continuation of a previous publication (Life - Basel 2022).
- General and specific
- The comparative evaluation Wt female/Tff3-female fed on standard diet should include gross dimensions of the intestinal segments as well as weight of the empty intestine.
- What abouti microvilli of Wt female/Tff-3 female fed on standard diet?
- How do the authors exclude the possibility that Tff3 deficiency does not induce behavioral changes in feeding?
- What about Fc Fragment of the IgG binding protein in the context of Tff3 silencing?
- Tff3 silencing: how do the authors eliminate an "off-target effect"?
- Why was 30 weeks chosen as the endpoint for evaluating the intestine? Additional data regarding the time-course of alterations (assessed at 21 weeks as in a previous article), as well as data regarding the reversibility of the modifications should be available.
- Quantitative evaluation of epithelial cell renewal in the various conditions should be shown.
- The discussion is too long. It includes statements that should be restricted to "results" .
- The histological technique is suboptimal.
- The observation of the reduction in brush border related to gender is insufficient. It must be accompanied by functional data such as measurements of brush-border enzymatic activities. in addition there is no real mechanistic insight.
- The authors should provide immunohistochemical evidence for Tff3 silencing in the intestine vs wild type.
- Text: Lieberkhün instead of Liberkin
- Table 1 should indicate number of determinations
- What is the translational relevance to human disease of the findings (celiac disease?)
some sentences are incomprehensible. Clarification of langage is desirable.
Author Response
Dear Mr. Rico Li, Assistant Editor and reviewer 1
Thank you for your letter and for the reviewers’ comments on our manuscript entitled " Tff3 deficiency differentially affects morphology of male and female intestine in long-term high-fat diet-fed mouse model (ID: ijms-2605601) and the opportunity given to revise our manuscript. We are also most grateful for granted deadline extension to submit the revised version of the manuscript.
The comments were very helpful for revising and improving our paper. We have considered comments raised by both reviewers and have revised the manuscript that we hope will meet your approval. The responses to the reviewers’ comments are provided in attached file.
We would like to express our greatest appreciation to you and the reviewers for the comments on our paper.
Kind regards,
Tatjana Pirman

Reviewer 2 Report
Comments and Suggestions for Authors
This is a well constructed and almost well-executed exploration of the effects of a trefoil-family protein broadly critical for metabolism and is found in places one might expect based on that function (e.g. GI tract, liver...). The researchers set out to examine the impacts of the deficiency using a new mouse model that cut down on background noise present in previous attempts at this analysis.
They conclude that, interestingly, there is a protective benefit to tff3-deficient male mice on a high fat diet, and some notable differences in crypt depth and affects between sexes. While the findings are not earth shattering, they are important for the record and worth publishing.
Much is made of the qPCR data, though as they note, they do not see downstream affects genes regulated by qPCR targets. They claim gender-neutral and significant difference in the expression of sxpb1, but these data set off a number of concerns for me.
1. Just zooming out, there is a concerning difference between the apparent quality of the data that tracks by graph bar color as opposed to any other pattern I can see. It, at a minimum, suggests perhaps a mistake during analysis and needs to be revisited to ensure the data is sound.
2. These fold changes are very small. I think these days it's ok to use qPCR for these kinds of modest changes, but if we're to do this, we must be extremely careful about the conclusions we draw from them, especially concerning their biological relevance.
3. Finally, if we're gonna support those biological relevancies we need to be careful of data quality which is the second reason to revisit all qPCR data to ensure nothing systemic.
Author Response
Dear Mr. Rico Li, Assistant Editor and reviewer 2
Thank you for your letter and for the reviewers’ comments on our manuscript entitled " Tff3 deficiency differentially affects morphology of male and female intestine in long-term high-fat diet-fed mouse model (ID: ijms-2605601) and the opportunity given to revise our manuscript. We are also most grateful for granted deadline extension to submit the revised version of the manuscript.
The comments were very helpful for revising and improving our paper. We have considered comments raised by both reviewers and have revised the manuscript that we hope will meet your approval. The responses to the reviewers’ comments are provided below.
We would like to express our greatest appreciation to you and the reviewers for the comments on our paper.
Kind regards,
Tatjana Pirman

Round 2
Reviewer 1 Report
Comments and Suggestions for Authors
The reviewer appreciates the effort made by the authors in addressing his questions and remarks. However, it is somewhat frustrating to see that several points remain unclear and require significant improvements.
"Female Tff3-/- mice (Figure 3c and 3d) showed a 50% reduction in microvilli height compared to that of female Wt mice (Figure 3c and 3d). This morphological and quantitative observation, which is the key element of the article, lacks statistical support in terms of number of animals subjected to ultrastructural examination, number of determinations performed, as well as their statistical significance: Figure S4 is incomplete.
The discussion in this article should address the following questions:
- why do the authors believe that the morphological changes of the brush border do not result from a behavioral change of the animals?
-
The authors should discuss the limitations of their study and suggest additional work that is necessary to confirm (enzymatic activities, etc.) and expand upon their morphological study
Finally, considering that this article is based on morphology, a graphical abstract should be included.
Comments on the Quality of English Language
- The text should be shortened
Author Response
Reviewer comments were addressed as follows:
Reviewers' comments:
Thank you for your comments and suggestions. We have considered them, revised the manuscript, and responded
"Female Tff3-/- mice (Figure 3c and 3d) showed a 50% reduction in microvilli height compared to that of female Wt mice (Figure 3c and 3d). This morphological and quantitative observation, which is the key element of the article, lacks statistical support in terms of number of animals subjected to ultrastructural examination, number of determinations performed, as well as their statistical significance: Figure S4 is incomplete.
The Figure S4 was changed with the graph with a new measurement that we made on Transmission electron microscopy (TEM) on the samples from both female groups. We analysed 3 animals per group, and each animal was presented with 4 different sections. In each section, the length of at least 10 different microvilli was measured. Unfortunately, we did not have any more the samples of males from this study and we are not able to add the additional results for male groups.
The fact is that, when we carried out measurements on histological samples (duodenum), we saw differences, especially in the samples of females of both groups. At maximum magnification of microscope, differences in microvilli length between groups were done. We know that those values are not precise, but it was seen the reduced microvilli length in the Tff3 female group (0.70 ± 0.03), compared to other groups (0.84 ± 0.04 WT females, 0.88 ± 0.06 Tff3 males and 0.96 ± 0.14 Wt males). Measurements were taken on all animals (6/group). When statistical was done, there was a statistically significant difference from Tff3 female group to both groups of males, while in females there was a statistically significant difference in the border (p = 0.0653).
The analysis TEM is very accurate, but also expensive, and at the beginning we have made pictures only for an individual animal per group, and we have seen that there is an actual difference between the two groups of females as well. Because of that the statistic was no able to do. No, after the additional analysis was made, 3 samples per group, statistically significant differences between these two groups were obtained, which we showed in the graph in supplement (Figure S3). We aware that those values are some different that we obtained in microscopy, but it is normal since the TEM is much more precise method then ordinary microscopy.
The discussion in this article should address the following questions:
- why do the authors believe that the morphological changes of the brush border do not result from a behavioral change of the animals?
-
- The authors should discuss the limitations of their study and suggest additional work that is necessary to confirm (enzymatic activities, etc.) and expand upon their morphological study
In the part of discussion was add the text below in which we add the answer to the questions, that reviewer wants from us to address:
“This manuscript presents part of the data from extensive research on the systemic impact of Tff3 deficiency on the gut-liver-brain axis. In concept of our research we have used prolonged high fat diet exposure to mimic physiologically relevant conditions including ageing contributing to different diseases in that axis. Here we present impact of Tff3 deficiency on intestine in both sexes. To our surprise the major intestinal finding was reduced surface of intestine in Tff3 deficient female mice on prolonged HFD (reduced duodenal villi height and microvilli length) at age of 40 weeks. This was not observed in Tff3-deficient female mice on standard diet of same age. During monitored period from 4 weeks up to end point at age of 40 weeks mice we performed functional metabolic tests of glucose and insulin tolerance presented elsewhere [46] we could see different time related changes but no major difference in body weight due to Tff3 deficiency. One of the possible reasons for noticed reduced intestinal surface in Tff3-deficient females on HFD could be related to feeding behavior or food absorption abilities of mice. Unfortunately, data collected by here presented research concept cannot resolve this issue. Additional experiments including enzymatic activities in brush border would be need to further address if feeding behavior is cause of noticed morphological change in Tff3-deficient female intestine on HFD.”
Round 3
Reviewer 1 Report
Comments and Suggestions for Authors
Figure S4 should be incorporated in the article and not be presented as a supplementary figure. Final version still needs english editing.
Comments on the Quality of English LanguageEnglish editing is necessary
Author Response
Reviewer comments were addressed as follows:
Reviewers' comments:
Thank you for your suggestions. We revised the manuscript, and responded.
Figure S4 should be incorporated in the article and not be presented as a supplementary figure.
The Figure S4 is now incorporated in the article as Figure 4 on page 5 and it is deleted in the supplementary file.
Final version still needs English editing.
The final version was sent to the English editing to AJE, the Editing Certificate is added in the sending material.

Round 4
Reviewer 1 Report
Comments and Suggestions for Authors
No comment
Comments on the Quality of English Languageno comment